# DGMS: Domain Generalization Method for Mamba-based Super-Resolution Networks

## Abstract

Mamba-based domain generalization methods can effectively improve the performance of Mamba networks on samples with unknown distributions. However, existing methods target high-level vision tasks like image and point cloud classification, with limited research on low-level vision tasks such as image super-resolution (SR). To bridge this gap, we propose a Domain Generalization method for Mamba-based Super-Resolution networks (DGMS), which introduces a domain shift metric for SR tasks and identifies key variables governing domain shifts. Subsequently, based on the identified key variables, we propose hidden state update regularization and parameter consistency regularization. Through explicit supervisory constraints on these key variables, the method effectively enhances the network's generalization performance across images with different degradation models. Extensive experiments across diverse data distributions and network architectures demonstrate the effectiveness of the proposed method on low-level vision tasks, where it outperforms existing state-of-the-art Mamba-based domain generalization methods. Our code is available at ***.

## 1 Introduction

Super-resolution (SR) networks have extensive applications, including remote sensing image enhancement (Xiao et al., 2024b; Kang et al., 2024), vintage video restoration (Tang et al., 2024; Zhou et al., 2024), image restoration (Wu et al., 2024b;a), and even serving as preprocessing for downstream tasks (Kim et al., 2024; Qiu et al., 2024). In SR tasks, expansive receptive fields enable more reference pixels during inference, significantly improving performance. Moreover, pixel-level prediction tasks like SR involve extremely long input sequences (equal to the number of image pixels). Due to their input-aware properties, Mamba networks offer larger receptive fields than CNNs while maintaining better computational efficiency than Transformers (Yu & Wang, 2024) (Mamba scales linearly with sequence length versus quadratic for Transformers). These advantages have motivated numerous Mamba-based SR networks (Guo et al., 2024b; Cheng et al., 2024a; Shi et al., 2025; Li et al., 2025; Guo et al., 2024a; Xiao et al., 2024a).

The promising performance of existing Mamba-based SR networks relies on a critical assumption: similar data distributions between the test (target domain) and training sets (source domain). However, in practical scenarios, domain shifts caused by varying illumination conditions and imaging devices lead to significant distribution discrepancies between source and target domains. This results in severe performance degradation when deploying source-domain-optimized networks to target domains, which is particularly detrimental for confidence-sensitive applications like medical imaging.

Domain generalization methods for Mamba networks have been explored in high-level vision tasks (e.g., image and point cloud classification) to improve robustness. These methods enable networks trained on source domains to maintain satisfactory performance on target domains with unknown distributions. Specifically, they decompose features into pixel-aligned foreground features (determining classification results) and background features (containing domain-specific information). As exemplified by (Long et al., 2024a) and (Huang et al., 2024a) which employ attribution algorithms and token affinity analysis respectively for feature separation. These methods then improve generalization by artificially perturbing background features: DGMamba (Long et al., 2024a) randomly permutes background token sequences, PointDGMamba (Yang et al., 2024) integrates features from

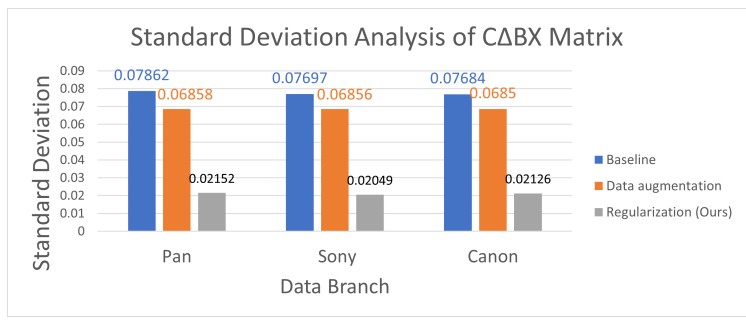

Figure 1: Analysis of $C\Delta BX$ Matrix Variation. We generate 600 LR image patch pairs with identical content but different degradation models under Pan, Sony, and Canon data branches. These patches are fed into networks employing transfer learning (Baseline), feature perturbation-based data augmentation, and our proposed regularization, respectively. We then analyze the standard deviation of $C\Delta BX$ matrix features under different degradation conditions. Higher values indicate greater sensitivity of the $C\Delta BX$ matrix to varying degradation models, leading to larger domain shifts and poorer network generalization performance.

same-class samples, while START (Guo et al., 2024c) and Huang et al. (2024a) perform style perturbation on background features using randomly generated samples.

However, domain generalization methods for Mamba-based SR networks remain unexplored. Direct application of existing methods would encounter two critical issues:

- Feature perturbation-based data augmentation methods impair SR network predictions. In high-level vision tasks that rely on abstract representations of entire images, modifications to background pixels (e.g., random permutation or replacement) do not affect the final prediction results. However, for pixel-level prediction tasks such as SR, perturbation methods involving random pixel rearrangement or replacement inevitably interfere with network predictions.

- Feature perturbation-based data augmentation methods demonstrate weak network constraints. As evidenced in Figure.1, these implicit methods that aim to reduce network sensitivity to specific features through perturbation demonstrate weaker feature constraints compared to explicitly constrained supervised training methods.

To address these two issues and inspired by the methods in (Pan et al., 2011; Ding et al., 2022; Guo et al., 2024c), we first introduce a domain shift metric specifically for Mamba-based SR networks to identify key variables governing domain shift. Subsequently, we propose a hidden state update regularization term and a parameter consistency regularization term to explicitly constrain these key variables. Specifically, the hidden state update regularization term constrains key variables controlling the retention degree of past hidden states, preventing the accumulation of domain shifts. The parameter consistency regularization term constructs LR images with identical content but different degradation models, enabling the network to maintain stability of key variables when processing LR images with varying degradation models. This effectively resolves both the interference of feature perturbation with network predictions and the insufficient constraint strength of feature perturbation on the SR network.

To the best of our knowledge, this work presents the first domain generalization study on Mamba networks for low-level vision tasks. Our main contributions are threefold:

- We introduce domain shift metric into low-level vision tasks by developing a dedicated metric for Mamba-based SR networks. Building upon this metric, we identify the key variables governing domain shifts.

- Building upon these identified key variables, we propose hidden state update regularization and parameter consistency regularization, which effectively enhance the generalization capability of Mamba networks through explicit supervisory constraints on these key variables.

- The proposed method exhibits plug-and-play functionality, being architecture-agnostic to various Mamba-based SR networks. Furthermore, it exhibits robust adaptability to diverse training distributions, outperforming existing state-of-the-art Mamba-based domain generalization methods on diverse test distributions.

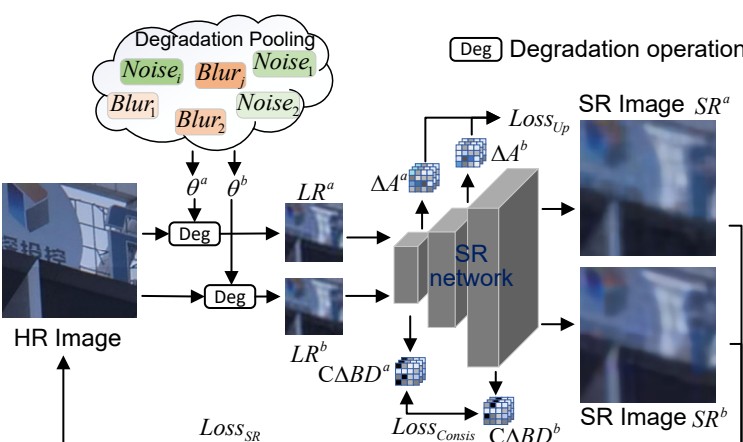

Figure 2: Schematic of Regularization. $Loss_{SR}$ denotes the pixel-wise loss for the SR network (L1 loss), $Loss_{Up}$ represents the hidden state update regularization loss, and $Loss_{Consis}$ corresponds to the parameter consistency regularization loss. $\theta$ is the degradation parameter randomly sampled from Degradation Pooling. $LR^a$ and $LR^b$ are degraded images generated from the HR image using different degradation parameters $\theta^a$ and $\theta^b$, which are then fed into the SR network to produce corresponding SR images $SR^a$ and $SR^b$. $\Delta A$ and $C\Delta BX$ are the obtained key variables.

Please refer to Appendix A for relevant work on Mamba and Domain Generalization.

## 2 METHOD

### 2.1 PRELIMINARY

Mamba networks achieve wide receptive fields with input-aware parameters while maintaining low computational overhead. As formalized in Eq. 1, the linear projection layer maps input $x$ to control matrices $B$, $C$, and $\Delta$. The step control parameter $\Delta$ then discretizes these matrices to ensure compatibility with neural network (Eq. 2),

$$B = S_B(x), C = S_C(x), \Delta = Softplus(S_\Delta(x)), \tag{1}$$

$$\overline{A} = e^{\Delta A}, \overline{B} = \Delta B, \tag{2}$$

where $Softplus(x) = log(1 + e^x)$, $S_B$, $S_C$, and $S_\Delta$ denote the linear projection layers for control matrices $B$, $C$, and $\Delta$ respectively. $\overline{A}$, $\overline{B}$ represent the discretized versions of matrices $A$ and $B$, and $A$ is a learnable parameter matrix.

As shown in Eq. 3, Mamba utilizes $\overline{A}$ and $\overline{B}$ to control the retention degree of previous hidden states and the influence of current inputs on current hidden states, respectively. When encountering representative inputs, Mamba preserves them in hidden states via elevated $\overline{B}$ values for subsequent inference. Conversely, insignificant inputs are forgotten through low $\overline{B}$ values. Combined with output control via $C$ matrix, this enables large receptive fields through historically significant information stored in hidden states.

$$h_t = \overline{A}_t h_{t-1} + \overline{B}_t x_t, \tag{3}$$

$$y_t = C_t h_t, \tag{4}$$

where $h_{t-1}$ and $h_t$ denote the hidden states at time steps $t-1$ and $t$ respectively, and $x_t$ represents the input at time step $t$.

For a Mamba-based SR network, the input $x \in \mathbb{R}^{B,C,H \times W}$ corresponds to the flattened features of the input image. The network iterates through features at each pixel position and processes them as $x_t \in \mathbb{R}^{B,C,1}$ inputs. The dimensions $B$, $C$, $H$, and $W$ denote the batch size, number of channels, height, and width, respectively. The output $Y \in \mathbb{R}^{B,C,H \times W}$ denotes the feature map produced by the Mamba backbone network. Subsequently, the SR network adjusts the feature dimensions and performs upsampling operations on $Y$ to generate the SR image $SR \in \mathbb{R}^{B,C,H \times S,W \times S}$, where $S$ denotes the upsampling scale factor.

## 2.2 THEORETICALLY ANALYSIS FOR THE DOMAIN SHIFT OF MAMBA

Following (Guo et al., 2024c), the domain shift between source and target domains can be formulated as:

$$|\overline{y}^s - \overline{y}^t| = \sum_{i=1}^{L} |\overline{y}_i^s - \overline{y}_i^t|, \tag{5}$$

where $L$ is the input sequence length, $\overline{y}^s$ and $\overline{y}^t$ denote the mean predictions over the source and target domains, respectively, while $\overline{y}_i^s$ and $\overline{y}_i^t$ correspond to the predictions at the $i-th$ timestep for source and target domains, respectively.

The upper bound of the increase in domain shift at time step $t$ can be expressed as:

$$|y_{i+1}^s - y_{i+1}^t| - |y_i^s - y_i^t|$$
$$\leq |(\frac{S_C(\overline{x}_{i+1}^s)}{S_C(\overline{x}_i^s)}e^{\tilde{S}_\Delta(\overline{x}_{i+1}^s)A} - 1)(y_i^s - y_i^t) + y_i^t[\frac{S_C(\overline{x}_{i+1}^s)}{S_C(\overline{x}_i^s)}e^{\tilde{S}_\Delta(\overline{x}_{i+1}^s)A} - \frac{S_C(\overline{x}_{i+1}^t)}{S_C(\overline{x}_i^t)}e^{\tilde{S}_\Delta(\overline{x}_{i+1}^t)A}]$$
$$+ (S_C(\overline{x}_{i+1}^s)\tilde{S}_\Delta(\overline{x}_{i+1}^s)S_B(\overline{x}_{i+1}^S)\overline{x}_{i+1}^S - S_C(\overline{x}_{i+1}^t)\tilde{S}_\Delta(\overline{x}_{i+1}^t)S_B(\overline{x}_{i+1}^t)\overline{x}_{i+1}^t)|, \tag{6}$$

where $\overline{x}_{i+1}$ and $\overline{x}_i$ denote the mean input features at step $i+1$ and step $i$, respectively. $\tilde{S}$ represents the $Softplus(S_\Delta)$ operation, and superscripts $s$ and $t$ correspond to the source domain and target domain, respectively.

The proof of the above theorem is provided in Appendix B. Through Eq.6, we derive the following two proposions:

- **Domain shift accumulation**: When Mamba networks sequentially scan features at each timestep of the input sequence, the domain shift gradually accumulates during the scanning process. (This phenomenon has been widely recognized by existing works (Long et al., 2024a; Huang et al., 2024a; Guo et al., 2024c))

- **Mamba networks for SR tasks exhibit more severe domain shift accumulation**: Unlike high-level vision tasks (e.g., image classification) where inputs are processed patch-wise (e.g., $14 \times 14$ patches $\rightarrow$ 196 timesteps in VMamba (Zhu et al., 2024)), Mamba based SR networks require processing each pixel sequentially, resulting in substantially longer sequences (e.g., $200 \times 200$ piex images $\rightarrow$ 40,000 timesteps in MambaIR (Guo et al., 2024b) and MaIR(Li et al., 2025)). Given the substantially longer sequence in SR tasks (approximately 204× those of image classification), Mamba networks for SR tasks demonstrate more pronounced domain shift accumulation compared to their high-level vision counterparts. Consequently, Mamba networks for low-level vision require stricter domain shift constraints compared to their high-level vision task counterparts.

## 2.3 KEY VARIABLE REGULARIZATION

As shown in Figure.2, we address domain shift accumulation through two components: (1) a hidden state update regularization term targeting the first two shift components (red and green terms in Eq. 6), and (2) a parameter consistency regularization term optimizing the final component (blue term in Eq. 6). The complete training pipeline is presented in Appendix C.

### 2.3.1 HIDDEN STATE UPDATE REGULARIZATION

The first two terms of the domain shift increment formula (Eq. 6) are decomposed as follows:

$$D1 = |(\frac{S_C(\overline{x}_{i+1}^s)}{S_C(\overline{x}_i^s)}e^{\tilde{S}_\Delta(\overline{x}_{i+1}^s)A} - 1)(y_i^s - y_i^t) + y_i^t[\frac{S_C(\overline{x}_{i+1}^s)}{S_C(\overline{x}_i^s)}e^{\tilde{S}_\Delta(\overline{x}_{i+1}^s)A} - \frac{S_C(\overline{x}_{i+1}^t)}{S_C(\overline{x}_i^t)}e^{\tilde{S}_\Delta(\overline{x}_{i+1}^t)A}]|.$$
(7)

From Eq. 7, it can be observed that $D1$ is primarily related to $\frac{S_C(\overline{x}_{i+1})}{S_C(\overline{x}_i)}$ and $e^{\tilde{S}_\Delta(\overline{x}_{i+1})A}$. Since it is difficult to achieve ultra-fine-grained control of $S_C(x)$ at the timestep level, and excessive control over features may restrict the network's flexibility in processing image features (As shown in Table.2), we focus on the term $e^{\tilde{S}_\Delta(\overline{x}_{i+1}^s)A}$ to impose constraints on domain shift. The decrease of $e^{\tilde{S}_\Delta(\overline{x}_{i+1}^s)A}$ can make the first term (red term) of the domain shift formula decrease. At the same time, it will also make the second term (green term) decrease because the subtracted terms decrease leading to the final difference value reduction. Therefore, we intend to add $e^{\tilde{S}_\Delta(\overline{x}_{i+1}^s)A}$ as a regularization term into the training process of the SR network. It should be noted that, since the positive-base exponential function does not change the variation characteristics of the exponent, after simplification $(\tilde{S}_\Delta(\overline{x}_{i+1}^s)A)$, it is added into the original SR network's loss function:

$$Loss_{All} = Loss_{SR}(SR, HR) + Loss_{Up}(\Delta A),$$
(8)

$$Loss_{Up}(\Delta A) = \frac{1}{N_A}\sum_{i=1}^{N_A}|\Delta_i A_i|,$$
(9)

where $Loss_{SR}$ denotes the base loss function of the SR network (typically L1 loss), $Loss_{Up}$ represents the hidden state update regularization term, and $N_A$ indicates the number of $\Delta A$ matrices involved in regularization.

**Empirical analysis**: In the previous section (Sction 2.2), we concluded that under the assumption where each element in the input sequence (image patches for high-level vision tasks, pixels for low-level vision tasks) contains certain domain shifts, due to Mamba's hidden state update and input-aware properties, the domain shifts in hidden states will gradually accumulate with increasing timesteps. In Mamba networks, the hidden state update is managed by two control matrices: $\overline{A}$ controls the retention degree of the previous timestep's hidden state, while $\overline{B}$ controls the influence degree of the current timestep's input. Domain shifts flow into hidden states through control matrix $\overline{B}$ along with input features, and gradually accumulate through control matrix $\overline{A}$. The proposed hidden state update regularization term constrains the control matrix $\overline{A}$ ($\overline{A} = e^{\tilde{S}_\Delta(\overline{x}_{i+1})A}$), attenuating the retention of previous timestep's hidden state, thereby preventing the accumulation of domain shifts.

### 2.3.2 PARAMETER CONSISTENCY REGULARIZATION

The final term of the domain shift increment formula (Eq. 6) are decomposed as follows:

$$D2 = |(S_C(\overline{x}_{i+1}^s)\tilde{S}_\Delta(\overline{x}_{i+1}^s)S_B(\overline{x}_{i+1}^S)\overline{x}_{i+1}^S - S_C(\overline{x}_{i+1}^t)\tilde{S}_\Delta(\overline{x}_{i+1}^t)S_B(\overline{x}_{i+1}^t)\overline{x}_{i+1}^t)|.$$
(10)

To reduce the $D2$ term, we aim to enhance the robustness of the $C\Delta Bx$ matrix to ensure consistent outputs for LR images with different degradation models. Existing domain generalization methods (Guo et al., 2024c), (Long et al., 2024a) typically apply perturbations to specific features to reduce the network's sensitivity to those features. As shown in Figure.1, when employing the style transfer method proposed in (Guo et al., 2024c)—combined with saliency guidance maps derived from $C\Delta BX$ to perturb features—the method only marginally reduces the sensitivity of the $C\Delta BX$ matrix to LR images under varying degradation models, compared to explicit constraint methods. Therefore, we propose to explicitly constrain the variation range of the $C\Delta BX$ matrix when processing LR images with different degradation models to reduce the final computed value of $D2$.

As illustrated in Figure.2, we propose a parameter consistency regularization term to explicitly constrain the variation range of the $C\Delta BX$ matrix for LR images under different degradation models.

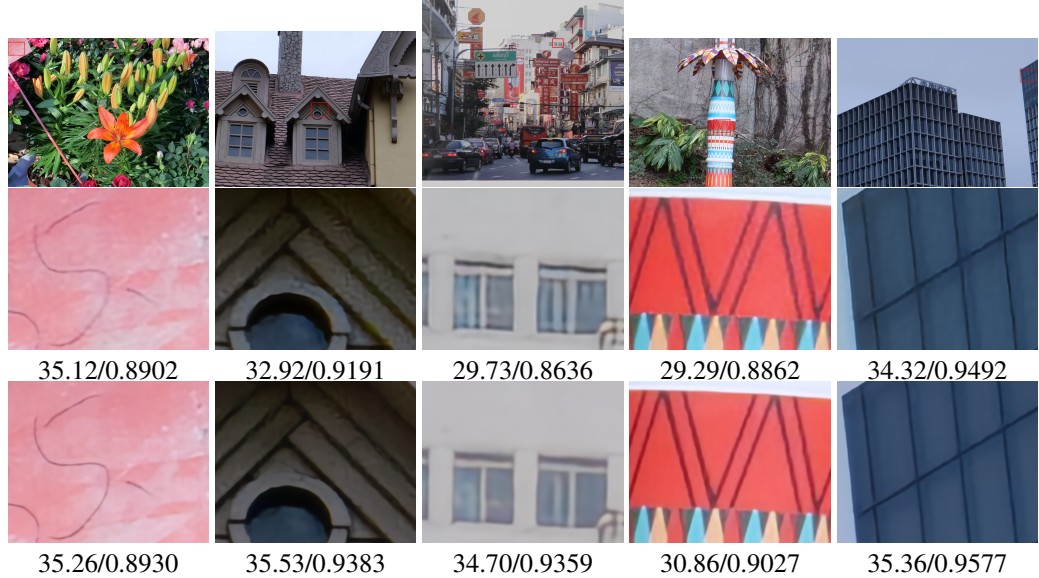

| 35.12/0.8902 | 32.92/0.9191 | 29.73/0.8636 | 29.29/0.8862 | 34.32/0.9492 |
| 35.26/0.8930 | 35.53/0.9383 | 34.70/0.9359 | 30.86/0.9027 | 35.36/0.9577 |

Figure 3: Visual comparison. Row 1: LR images. Row 2: Super-resolved results by MambaIR(Guo et al., 2024b). Row 3: Super-resolved results by MambaIR+DGMS. Numerical values below images represent PSNR/SSIM metrics. (Please view zoomed-in on screen.)

Specifically, we randomly sample two distinct degradation parameters $\theta^a$ and $\theta^b$ from the degradation pool to generate a pair of LR images with identical content but different degradation models,

$$\theta^a, \theta^b = Random\_sampling(Degradation\_pool), \tag{11}$$

$$LR^a = De(HR, \theta^a), LR^b = De(HR, \theta^b), \tag{12}$$

where $De$ employs degradation parameters $\theta$ to degrade the $HR$ image, generating an $LR$ image.

The two LR images with distinct degradation models are fed into the SR network to extract their respective SR outputs, $\Delta A$ matrices, and $C\Delta BX$ matrices,

$$SR^a, \Delta^a A^a, C\Delta BX^a = M(LR^a),$$
$$SR^b, \Delta^b A^b, C\Delta BX^b = M(LR^b). \tag{13}$$

Building upon the hidden state update regularization term proposed in the previous subsection, we incorporate the parameter consistency regularization term $Loss_{Consis}$ into the overall loss function,

$$Loss_{All} = Loss_{SR}(SR^a, HR) + Loss_{SR}(SR^b, HR) + Loss_{Up}(\Delta^a A^a) + Loss_{Up}(\Delta^b A^b)$$
$$+ Loss_{Consis}(C\Delta BX^a, C\Delta BX^b) + Loss_{Consis}(C\Delta BX^b, C\Delta BX^a). \tag{14}$$

Inspired by (Wang et al., 2024b), we recognize that overly stringent constraints on output features may compromise network performance. Thus, we restrict the computation of $C\Delta BX$ matrix feature differences to the channel dimension exclusively,

$$Loss_{Consis}(f^a, f^b) = ||Cov(f^a) - Cov(f^b)||^2 + ||\mu(f^a) - \mu(f^b)||^2$$
$$+ ||Cov(hx^a) - Cov(hx^b)||^2 + ||\mu(hx^a) - \mu(hx^b)||^2, \tag{15}$$

where, $f^a$ is obtained by performing channel-wise average pooling on $C\Delta BX^a$. The term $hx^a$ represents the features of $C\Delta BX^a$ in the reproducing kernel Hilbert space (RKHS), with the projection function defined as:$H = \{h : \sqrt{2}cos(\omega x + \phi)|\omega \sim \mathcal{N}(0, 1), \phi \sim U(0, 2\pi)\}$. $\mu$ and $Cov$ denote the mean and covariance matrix, respectively. The variables $f^b$ and $hx^b$ are derived analogously to $f^a$ and $hx^a$.

Table 1: Effectiveness validation of DGMS. The source domain comprises the Olympus and DSC data branches, with PSNR/SSIM adopted as the performance metrics. The best performance is bolded. **The experiment was repeated four times, with results reported as mean $\pm$ standard deviation.**

| Method | Pan | Sony |
|---|---|---|
| Baseline | 30.81/**0.8688** | 30.81/0.8850 |
| +HUR | $31.26 \pm 0.017$/$0.8627 \pm 1.73$e-4 | $31.39 \pm 0.025$/$0.8796 \pm 3.79$e-4 |
| + HUR + PCR | **31.37 $\pm$ 0.021**/$0.8635 \pm 2.71$e-4 | **31.81 $\pm$ 0.005**/**0.8854 $\pm$ 8.16e-5** |

| Method | Canon | |
|---|---|---|
| Baseline | 30.93/0.8617 | |
| +HUR | $32.51 \pm 0.060$/$0.9257 \pm 2.31$e-4 | |
| + HUR + PCR | **33.19 $\pm$ 0.042**/**0.9279** $\pm$ 2.83e-4 | |

Table 2: Ablation experiments on different control matrices

| Method | Pan | Sony | Canon |
|---|---|---|---|
| $A$ | 31.06/0.8615 | 30.96/0.8680 | 32.24/0.9237 |
| $\Delta$ | 31.17/0.8620 | 31.24/0.8772 | 32.18/0.9233 |
| $\Delta A + Var(C)$ | 31.02/0.8587 | 31.11/0.8743 | 32.44/**0.9261** |
| $\Delta A$ | **31.26/0.8627** | **31.39/0.8796** | **32.51**/0.9257 |

## 3 EXPERIMENT

### 3.1 EXPERIMENTAL DETAILS

The DRealSR (Wei et al., 2020), Set5 (Bevilacqua et al., 2012), Set14 (Zeyde et al., 2010), B100 (Martin et al., 2001), Urban (Huang et al., 2015), Manga109 (Matsui et al., 2017), and DIV2K (Timofte et al., 2017) datasets were employed to validate the effectiveness of the proposed DGMS method. The DRealSR dataset comprises real-world images captured by diverse cameras (e.g., Panasonic, Sony, Olympus, Canon, and DSC), each introducing unique degradation characteristics. To evaluate the generalization capability of our method across diverse data distributions, we adopt a cross-sample-branch evaluation protocol: images from one sample branch serve as the training set, while those from the remaining sample branches constitute the test set. Furthermore, we deploy the proposed method on diverse network architectures (MambaIR (Guo et al., 2024b), MMA (Cheng et al., 2024a), and Mair(Li et al., 2025)) to verify its architecture-agnostic generalization. All experiments are conducted on $4 \times$ Tesla V100 GPUs, with a patch size of $48 \times 48$, a batch size of 16, and the Adam optimizer.

### 3.2 ABLATION EXPERIMENT

The ablation studies on (1) different data distributions, (2) different hidden state update regularization layers and (3) different parameter consistency regularization layers are presented in Appendix D.

#### 3.2.1 EFFECTIVENESS ABLATION EXPERIMENTS FOR EACH COMPONENT OF DGMS

To validate the effectiveness of the proposed modules, ablation experiments were conducted for each component. The transfer learning method on the Olympus data branch serves as the baseline. As demonstrated in Table.1, the introduction of the hidden state update regularization term (HUR) mitigates the negative impact of domain shift on network performance, achieving a 0.45 dB PSNR improvement on the Pan data branch. Furthermore, the parameter consistency regularization term (PCR) enhances the robustness of the $C\Delta BX$ matrix to LR images with varying degradation models, further improving the generalization capability of the SR network. Compared to the baseline, the proposed method delivers a 0.55 dB PSNR gain, validating its efficacy. As shown in Figure.3, the MambaIR (Guo et al., 2024b) network with the proposed DGMS regularization yields sharper image edges while effectively suppressing noise.

Table 3: Ablation experiments on different feature perturbation methods

| Method | Pan | Sony | Canon |
|---|---|---|---|
| PCR - $C\Delta BX$ | 31.16/0.8604 | 31.42/0.8788 | 32.79/0.9270 |
| Global feature | 31.23/0.8613 | 31.46/0.8799 | 32.82/0.9274 |
| Stylization | 31.28/0.8609 | 31.41/0.8774 | 32.70/0.9261 |
| Ours | **31.37/0.8635** | **31.81/0.8854** | **33.19/0.9279** |

Table 4: Ablation experiments on different network architectures

| Method | Pan | Sony | Canon |
|---|---|---|---|
| MMA (Cheng et al., 2024a) | 30.30/0.8564 | 30.39/0.8634 | 31.03/0.9174 |
| +DGMS | 30.82/0.8535 | 31.35/0.8785 | 32.55/0.9240 |
| MambaIR (Guo et al., 2024b) | 30.81/0.8688 | 30.81/0.8850 | 30.93/0.8617 |
| +DGMS | 31.37/0.8635 | 31.81/0.8854 | 33.19/0.9279 |
| Mair (Li et al., 2025) | 31.05/0.8626 | 30.88/0.8720 | 31.99/0.9220 |
| +DGMS | 31.18/0.8617 | 30.81/0.8676 | 32.35/0.9230 |

### 3.2.2 ABLATION EXPERIMENTS ON HIDDEN STATE UPDATE REGULARIZATION

As shown in Table.2, we investigate the impact of different control matrix selections in the hidden state update regularization term on network performance. While applying regularization constraints to individual components ( $\Delta$ or $A$) can partially restrict the variation range of the composite term ($\Delta A$), this method fails to guarantee global optimality. Compared to joint constraint, regularizing only $A$ or $\Delta$ alone leads to performance degradation of 0.2 dB and 0.09 dB in PSNR (Pan data branch), respectively. Additionally, we examine the effect of constraining matrix $C$. After implementing the $\Delta A$ matrix constraint, we introduce a variance regularization term for $C$ to reduce its variation range, thereby controlling the magnitude of $\frac{C_{i+1}}{C_i}$. However, this additional constraint results in a 0.24 dB PSNR drop on the Pan data branch. Excessive constraints on the control matrix $C$ will compromise its flexibility when processing varying LR images.

### 3.2.3 ABLATION EXPERIMENTS ON DIFFERENT FEATURE PERTURBATION METHODS

To investigate the impact of different feature perturbation methods in parameter consistency regularization on network generalization, we implemented existing Mamba network domain generalization methods for SR tasks. First, we analyzed the performance impact of removing the $C\Delta BX$ constraint, maintaining only the SR loss and hidden state update regularization when processing images with varying degradation models (PCR - $C\Delta BX$). Table.3 reveals that eliminating the $C\Delta BX$ constraint causes a 0.21 dB PSNR decline on the Pan data branch, quantitatively validating its importance for SR network generalization. We then implemented a global feature constraint analogous to (Wang et al., 2024b), operating on the backbone's final layer outputs rather than $C\Delta BX$ features (Global feature). This method underperforms the $C\Delta BX$ -constrained method by 0.13 dB PSNR (Pan branch, Table.3). To verify the impact of feature stylization perturbation on network robustness, a method similar to (Guo et al., 2024c) was adopted (Stylization). First, the $C\Delta BX$ feature matrix was calculated, features with values in the bottom half were selected as background feature, and their feature styles were randomly perturbed (similar to AdaIN (Huang & Belongie, 2017)). As shown in Figure.1, compared with explicit constraints, the perturbation produced weaker domain-invariant characteristics, resulting in lower generalization performance than the explicit constraint method. Compared with the explicit constraint method, the style perturbation-based method showed a 0.47 dB decrease in PSNR performance on the Canon data branch.

### 3.2.4 ABLATION EXPERIMENTS ON DIFFERENT NETWORK ARCHITECTURES

To validate the generalization capability of the proposed method across different network architectures, we implemented our method, DGMS, on three distinct networks: MMA (Cheng et al., 2024a), MambaIR (Guo et al., 2024b), and MaIR (Li et al., 2025). As evidenced by Table.4, the proposed method demonstrates robust generalization performance across different architectures. Notably, when equipped with DGMS, MambaIR achieves a 2.29 dB PSNR improvement on the Canon data branch.

Table 5: Comparative experiments. - indicates that the current method is inapplicable to the SR task. None denotes that the data was not provided in the original paper. DA denotes domain adaptation methods, AG refers to data augmentation methods, and DG represents domain generalization methods. Note that DA requires both source-domain and target-domain samples for domain-adaptive training, whereas DG only needs source-domain samples for training. The performance metrics consist of PSNR/SSIM/LPIPS. The best and second best performance are in red and blue colors, respectively.

| Method | DG&DA&AG | Pan | Sony | Canon |
|---|---|---|---|---|
| ZSSR (Shocher et al., 2018) | DA | 30.43/0.8434/0.4124 | 31.32/0.8712/0.3276 | 31.89/0.9134/0.3040 |
| IODA (Tang & Yang, 2024) | DA | 30.90/0.8594/0.3531 | 31.31/0.8807/0.3211 | 31.87/0.9216/0.2711 |
| SRTTA (Deng et al., 2023) | DA | 29.88/0.8359/0.4561 | 31.24/0.8714/0.3718 | 31.88/0.9146/0.3323 |
| DADA (Xu et al., 2022) | DA | 31.27/0.824/None | 32.05/0.843/None | None |
| Wang et al. (2024b) | AG | 31.28/0.8626/0.3306 | 31.53/0.8818/0.2909 | 32.72/0.9269/0.2630 |
| DGMamba (Long et al., 2024a) | DG | - | - | - |
| PointDGMamba (Yang et al., 2024) | DG | - | - | - |
| DTAM (Huang et al., 2024a) | DG | 31.23/0.8615/0.3350 | 31.29/0.8773/0.2915 | 32.65/0.9256/0.2657 |
| START (Guo et al., 2024c) | DG | 29.88/0.8359/0.3502 | 31.24/0.8714/0.3130 | 31.88/0.9146/0.2841 |
| DGMS | DG | 31.37/0.8635/0.3367 | 31.81/0.8854/0.2883 | 33.19/0.9279/0.2556 |

## 3.3 COMPARATIVE EXPERIMENTS

As shown in Table.5, we compare the proposed method (DGMS) with other domain generalization methods for Mamba networks, including DGMamba (Long et al., 2024a), PointDGMamba(Yang et al., 2024), DTAM (Huang et al., 2024a), and START(Guo et al., 2024c). Among them, DG-Mamba is difficult to apply to SR tasks because its attribution algorithm is challenging to implement for SR. PointDGMamba requires target class information for feature merging, but such category information is typically hard to obtain in SR tasks. Additionally, we compare with Wang et al. (2024b), a data augmentation-based domain generalization method, as well as domain adaptation methods that require target domain samples during training (ZSSR (Shocher et al., 2018), IODA (Tang & Yang, 2024), DADA(Xu et al., 2022), SRTTA(Deng et al., 2023)). Refer to Appendix E for visual comparisons.

## 4 CONCLUSION

In this paper, we propose a domain generalization method for Mamba-based SR networks (DGMS), which effectively enhances the performance of Mamba-based SR networks on target domain samples with unknown distributions after training with source domain samples. DGMS first introduces domain shift metric into SR tasks, identifying key variables that dominate domain shift. For these identified key variables, DGMS proposes both a hidden state update regularization term and a parameter consistency regularization term, effectively addressing two critical issues in existing methods: feature perturbations interfering with network predictions, and insufficient constraints for SR networks. The DGMS method outperforms state-of-the-art domain generalization methods for Mamba networks, offering novel insights for developing Mamba-based generalization methods for low-level vision tasks.

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

## A RELATED WORKS

### A.1 MAMBA

#### A.1.1 MAMBA FOR OTHER TASKS

Due to its input-aware properties, Mamba networks (Gu & Dao, 2023) demonstrated both a larger receptive field compared to convolutional networks and superior computational efficiency relative to Transformers (Vaswani et al., 2017). Specifically, Mamba exhibited linear complexity with respect to input length, while Transformers scaled quadratically (Yu & Wang, 2024). These advantages prompted the development of numerous Mamba variants for visual tasks. Zhu et al. (2024) adapted Mamba networks (Gu & Dao, 2023) to vision tasks by proposing a bidirectional scanning strategy, which resolved the inability of unidirectional scanning to capture contextual information. Recognizing that Fourier-transformed frequency-domain features exhibit position-invariant properties while retaining global image information, Zhang et al. (2024) incorporated these spectral features into Mamba to assist network inference. Hu et al. (2024) observed that existing scanning methods dispersed spatially adjacent pixels, and consequently developed a zigzag scanning algorithm that maintained spatial continuity while reducing redundancy through unidirectional passes. Noting

that manually-designed scanning strategies still lacked spatial awareness, Xiao et al. (2024c) introduced graph-based scanning, enabling networks to autonomously determine scan paths via minimum spanning trees. Chen et al. (2025) proposed a hierarchical Mamba architecture that employed three distinct network branches to extract multi-granularity features—coarse-level features captured global scene context, while fine-level features preserved local details. He et al. (2024) introduced a lightweight Mamba variant that partitioned input features channel-wise into three streams processed via: wavelet transforms, multi-scale convolutions, and residual connections, achieving competitive accuracy with reduced complexity. Wang et al. (2024a) identified that Mamba networks suffered from performance-degrading artifact noise. They addressed this by inserting multiple registration blocks into the input features and utilizing these blocks for inference prediction, effectively reducing artifact interference during network prediction. Hatamizadeh & Kautz (2024) pioneered a hybrid architecture combining Mamba and Transformer blocks, demonstrating superior performance on vision tasks.

### A.1.2 MAMBA FOR SR TASKS

Due to the exceptionally long input sequence in SR networks (typically equal to the number of image pixels) and the high demand for large receptive fields, numerous Mamba-based SR networks emerged.

MambaIR (Guo et al., 2024b) first introduced Mamba networks to SR tasks. Subsequently, Cheng et al. (2024a) incorporated channel attention mechanisms into the bidirectional scanning framework to enhance feature discrimination. Shi et al. (2025) extended existing four-directional scanning by implementing additional channel-wise feature scanning. Ren et al. Ren et al. (2024) proposed both hierarchical and vertical-horizontal sampling strategies, which significantly improved network performance while effectively reducing computational costs. Li et al. (2025) proposed a sliding band scanning strategy, where images were partitioned into multiple bands that were processed sequentially (i.e., subsequent bands were scanned only after completing the current band). Furthermore, these bands were progressively shifted across network layers to prevent pixels from persistently residing at band edges. Guo et al. (2024a) extended the work of (Guo et al., 2024b), arguing that the multi-directional scan in (Guo et al., 2024b) led to severe feature redundancy. To address this, it introduced a semantic attention mechanism, guiding feature scans in an instance-by-instance manner based on learned attention maps. Xiao et al. (2024a) incorporated Fast Fourier Transform (FFT) into Mamba blocks, leveraging frequency information to enhance SR inference. Lei et al. (2024) focused on lightweighting MambaSR networks, employing large-network distillation to compress Mamba-based SR networks. Di et al. (2024) first applied Mamba networks to Burst SR, noting that existing image alignment methods processed auxiliary and base frames pairwise, lacking global context. Their proposed QMambaBSR treated the base frame as a Query to aggregate information from all auxiliary frames. Huang et al. (2024b) pioneered Mamba for infrared image SR, extracting multi-receptive-field features via varied kernel sizes and further expanding the receptive field with wavelet transforms. Ji et al. (2024) were the first to introduce Mamba blocks for MRI image SR, while Lu et al. (2024) adapted Mamba for light-field images by proposing cross-merge scanning to handle high-dimensional light-field features. Gao et al. (2024) proposed a cross-view scanning strategy to address inter-view feature interference during scanning in existing light-field SR methods.

## A.2 DOMAIN GENERALIZATION

### A.2.1 DOMAIN GENERALIZATION FOR OTHER NETWORK ARCHITECTURES

Due to variations in imaging devices and capture environments, the distribution of test samples (target domain) often diverged from that of training samples (source domain), a phenomenon commonly termed domain shift. This shift caused networks that performed well on the source domain to suffer significant performance degradation on the target domain. Domain generalization methods aimed to mitigate this issue by either leveraging data augmentation or implicitly learning domain-invariant and domain-specific representations during training, thereby enhancing model robustness and ensuring consistent performance across different domains.

Cheng et al. (2024b) integrated large language models (LLMs) into the training process, leveraging their ability to describe domain-related and invariant characteristics to guide subsequent network classification. Guo et al. (2023) separated domain-invariant and domain-specific features at the

channel level, applying Dropout perturbation to domain-specific channels to enhance model robustness. However, Wang et al. (2024b) later demonstrated that Dropout could potentially compromise feature diversity. Long et al. (2024b) identified two limitations in existing methods: inter-channel feature redundancy and the coarse granularity of class-wise domain alignment techniques. They addressed these by first identifying redundant channels through activation divergence between source and target domains, then employing a multi-expert system for finer-grained domain representation. Hu et al. (2023) argued that prior domain generalization methods overemphasized domain-invariant feature extraction (i.e., compressing all domain features into a shared space), which risked discarding valuable domain-specific information. Their proposed DandelionNet preserved domain-invariant features while actively utilizing domain-specific characteristics. Li et al. (2024) leveraged text features from CLIP (Radford et al., 2021) to guide mask generation, constraining the original network's features to focus on specific spatial regions and channel-wise characteristics, thereby enhancing robustness. Vidit et al. (2023) pioneered the application of domain generalization to object detection, utilizing textual descriptions of domain differences and CLIP-guided feature adaptation. Cai et al. (2024) proposed continuous temporal domain generalization, grounded in the assumptions of data evolution continuity and network adaptation continuity. Yu & Hwang (2024) improved generalization by dynamically assigning test samples to dedicated domain experts. Zhao et al. (2024) augmented feature diversity through mixed point cloud representations, combining spatial, intensity, and semantic modalities. Pahk et al. (2025) decoupled the backbone network from task-specific heads during training, preventing interference from randomly initialized heads on well-pretrained features, which significantly boosted generalization performance.

### A.2.2 DOMAIN GENERALIZATION FOR MAMBA

Current domain generalization methods for Mamba networks targeted high-level vision tasks such as image classification and point cloud classification. These methods typically partitioned features into foreground features (corresponding to pixels) and background features, where foreground features determined the network's classification predictions, while background features contained domain-specific information. Subsequently, these methods introduced perturbations to the background features to enhance the network's robustness against domain-specific variations.

Long et al. (2024a)first employed attribution algorithms to identify foreground and background features corresponding to pixels, then replaced background features with those from other samples in the same batch and performed random permutation operations. Huang et al. (2024a) partitioned features according to inter-token affinity, designating high-affinity tokens as foreground features and low-affinity tokens as background features. The background features then underwent style-swapping perturbations with random sequences to enhance network robustness against them. Guo et al. (2024c) proposed the START method which divided features according to their activation values, based on the observation that features influencing network decisions typically exhibited higher activations. Specifically, features in the top 50% activation range were treated as foreground, while the remainder were considered background. The background features were then perturbed using methods similar to (Huang et al., 2024a). Yang et al. (2024) employed neural networks to implicitly classify foreground and background features. They processed point cloud features of the same category through mutual perturbation, enabling the network to automatically distinguish foreground-background features while filtering background point cloud noise.

The aforementioned domain generalization methods for Mamba networks targeted high-level vision tasks. These methods typically decomposed features into foreground features (determining network predictions) and background features (containing domain-specific information). Existing methods enhanced network robustness by perturbing background features in images, thereby improving generalization performance on unseen target domain distributions. In high-level vision tasks, the extracted features represented abstract of entire images, where local pixel variations rarely affected the global semantic interpretation. However, for pixel-level prediction tasks like SR - which required precise inference for every pixel - any pixel-level modification (e.g., random pixel permutations in DGMamba (Long et al., 2024a) ) significantly impacted prediction quality. In addition, compared to explicit constraints that directly enforced consistency across different degradation models, implicit constraint methods (achieved through feature perturbation) imposed weaker regularization on the SR network. Therefore, we proposed a domain generalization method specifically designed for Mamba-based SR networks. Our method introduced domain shift metric into the SR framework to identify key variables governing domain shifts. Based on these identified variables, we developed

two novel regularization terms: (1) a hidden state update regularization term and (2) a parameter consistency regularization term. These terms constrained the network to maintain consistency in the key variables when processing images from different degradation models. Compared to feature perturbation-based methods, our method demonstrated superior generalization performance on target domain samples with unknown distributions.

## B THEORETICAL PROOFS

**Theorem.** Following (Guo et al., 2024c), the domain shift between source and target domains can be formulated as:

$$|\overline{y}^s - \overline{y}^t| = \sum_{i=1}^{L} |\overline{y}_i^s - \overline{y}_i^t|, \tag{16}$$

where $L$ is the input sequence length, $\overline{y}^s$ and $\overline{y}^t$ denote the mean predictions over the source and target domains, respectively, while $\overline{y}_i^s$ and $\overline{y}_i^t$ correspond to the predictions at the $i-th$ timestep for source and target domains, respectively.

The upper bound of the increase in domain shift at time step $t$ can be expressed as:

$$|y_{i+1}^s - y_{i+1}^t| - |y_i^s - y_i^t|$$
$$\leq |(\frac{S_C(\overline{x}_{i+1}^s)}{S_C(\overline{x}_i^s)}e^{\tilde{S}_\Delta(\overline{x}_{i+1}^s)A} - 1)(y_i^s - y_i^t) + y_i^t[\frac{S_C(\overline{x}_{i+1}^s)}{S_C(\overline{x}_i^s)}e^{\tilde{S}_\Delta(\overline{x}_{i+1}^s)A} - \frac{S_C(\overline{x}_{i+1}^t)}{S_C(\overline{x}_i^t)}e^{\tilde{S}_\Delta(\overline{x}_{i+1}^t)A}]$$
$$+ (S_C(\overline{x}_{i+1}^s)\tilde{S}_\Delta(\overline{x}_{i+1}^s)S_B(\overline{x}_{i+1}^S)\overline{x}_{i+1}^S - S_C(\overline{x}_{i+1}^t)\tilde{S}_\Delta(\overline{x}_{i+1}^t)S_B(\overline{x}_{i+1}^t)\overline{x}_{i+1}^t)|, \tag{17}$$

where $\overline{x}_{i+1}$ and $\overline{x}_i$ denote the mean input features at step $i+1$ and step $i$, respectively. $\tilde{S}$ represents the $Softplus(S_\Delta)$ operation, and superscripts $s$ and $t$ correspond to the source domain and target domain, respectively.

**Proof.**

The inference process of the Mamba network is presented below (corresponding to Eq.3 and 4 in the main paper):

$$h_t = \overline{A}_t h_{t-1} + \overline{B}_t x_t, \tag{18}$$

$$y_t = C_t h_t, \tag{19}$$

where $h_{t-1}$ and $h_t$ denote the hidden states at time steps $t-1$ and $t$ respectively, and $x_t$ represents the input at time step $t$, and $\overline{A}, \overline{B}, C$ are the control matrices.

Expanding Eq.18 and 19, the output at the $i-th$ time step can be expressed as:

$$y_i = C_i \prod_{k=2}^{i} \overline{A}_k \overline{B}_1 x_1 + C_i \prod_{k=3}^{i} \overline{A}_k \overline{B}_2 x_2 + \cdots C_i \overline{B}_i x_i. \tag{20}$$

Under unified representation, the output at the $i-th$ timestep is: $y_i = \sum_{j=1}^{i} \alpha_{i,j} x_j$, where $\alpha_{i,j}$ is given by:

$$\alpha_{i,j} = C_i \prod_{k=j+1}^{i} \overline{A}_k \overline{B}_j \quad st. \quad 0 \leq j \leq i \leq L. \tag{21}$$

Based on Eq.16 and Eq.21, the domain shift can be further decomposed into timestep-wise components as:

$$|\overline{y}_i^s - \overline{y}_i^t| = |\sum_{j=1}^{i}(\alpha_{i,j}^s \overline{x}_j^s - \alpha_{i,j}^t \overline{x}_j^t)|, \tag{22}$$

where $\overline{x}_j^s$ and $\overline{x}_j^t$ denote the mean feature values at the $j-th$ timestep for the source and target domains, respectively.

Thus, based on Eq. 22, the upper bound of the increase in domain shift between timesteps $i+1$ and $i$ can be formulated via Eq. 23,

$$|y_{i+1}^s - y_{i+1}^t| - |y_i^s - y_i^t| = |\sum_{j=1}^{i+1}(\alpha_{i+1,j}^s \overline{x}_j^s - \alpha_{i+1,j}^t \overline{x}_j^t)| - |\sum_{j=1}^{i}(\alpha_{i,j}^s \overline{x}_j^s - \alpha_{i,j}^t \overline{x}_j^t)|$$

$$\leq |\sum_{j=1}^{i}[(\alpha_{i+1,j}^s - \alpha_{i,j}^s)\overline{x}_j^s - (\alpha_{i+1,j}^t - \alpha_{i,j}^t)\overline{x}_j^t] + (\alpha_{i+1,i+1}^s \overline{x}_{i+1}^s - \alpha_{i+1,i+1}^t \overline{x}_{i+1}^t)|, \quad (23)$$

where the derivation of the formula invokes the inequality $|x| - |y| \leq |x - y|$.

From Eq. 21, we derive:(by substituting Eq.1 and Eq.2 in the main paper)

$$\alpha_{i+1,j} - \alpha_{i,j} = C_{i+1} \prod_{k=j+1}^{i+1} \overline{A}_k \overline{B}_j - C_i \prod_{k=j+1}^{i} \overline{A}_k \overline{B}_j$$

$$= (\frac{C_{i+1}}{C_i}\overline{A}_{i+1} - 1)\alpha_{i,j} \quad (24)$$

$$= [\frac{S_C(\overline{x}_{i+1})}{S_C(\overline{x}_i)}e^{\tilde{S}_\Delta(\overline{x}_{i+1})A} - 1]\alpha_{i,j},$$

where, $\tilde{S}$ denotes the $Softplus(S_\Delta)$ operation.

Substituting Eq.24 into Eq.23, the upper bound of the increase in domain shift can be expressed as:

$$|y_{i+1}^s - y_{i+1}^t| - |y_i^s - y_i^t|$$

$$\leq |\sum_{j=1}^{i}[((\frac{S_C(\overline{x}_{i+1}^s)}{S_C(\overline{x}_i^s)}e^{\tilde{S}_\Delta(\overline{x}_{i+1}^s)A} - 1)\alpha_{i,j})\overline{x}_j^s - ((\frac{S_C(\overline{x}_{i+1}^t)}{S_C(\overline{x}_i^t)}e^{\tilde{S}_\Delta(\overline{x}_{i+1}^t)A} - 1)\alpha_{i,j})\overline{x}_j^t] +$$

$$(\alpha_{i+1,i+1}^s \overline{x}_{i+1}^s - \alpha_{i+1,i+1}^t \overline{x}_{i+1}^t)| \quad (25)$$

$$= |(\frac{S_C(\overline{x}_{i+1}^s)}{S_C(\overline{x}_i^s)}e^{\tilde{S}_\Delta(\overline{x}_{i+1}^s)A} - 1)y_i^s - (\frac{S_C(\overline{x}_{i+1}^t)}{S_C(\overline{x}_i^t)}e^{\tilde{S}_\Delta(\overline{x}_{i+1}^t)A} - 1)y_i^t$$

$$+ S_C(\overline{x}_{i+1}^s)\tilde{S}_\Delta(\overline{x}_{i+1}^s)S_B(\overline{x}_{i+1}^S)\overline{x}_{i+1}^S - S_C(\overline{x}_{i+1}^t)\tilde{S}_\Delta(\overline{x}_{i+1}^t)S_B(\overline{x}_{i+1}^t)\overline{x}_{i+1}^t|.$$

Further simplifying Eq.25, the upper bound of the domain shift increase decomposes into three components, labeled in red, green, and blue respectively,

$$|y_{i+1}^s - y_{i+1}^t| - |y_i^s - y_i^t|$$

$$\leq |(\frac{S_C(\overline{x}_{i+1}^s)}{S_C(\overline{x}_i^s)}e^{\tilde{S}_\Delta(\overline{x}_{i+1}^s)A} - 1)(y_i^s - y_i^t) + y_i^t[\frac{S_C(\overline{x}_{i+1}^s)}{S_C(\overline{x}_i^s)}e^{\tilde{S}_\Delta(\overline{x}_{i+1}^s)A} - \frac{S_C(\overline{x}_{i+1}^t)}{S_C(\overline{x}_i^t)}e^{\tilde{S}_\Delta(\overline{x}_{i+1}^t)A}]$$

$$+ (S_C(\overline{x}_{i+1}^s)\tilde{S}_\Delta(\overline{x}_{i+1}^s)S_B(\overline{x}_{i+1}^S)\overline{x}_{i+1}^S - S_C(\overline{x}_{i+1}^t)\tilde{S}_\Delta(\overline{x}_{i+1}^t)S_B(\overline{x}_{i+1}^t)\overline{x}_{i+1}^t|. \quad (26)$$

## C    THE PSEUDOCODE OF THE OVERALL TRAINING PIPELINE

The overall training framework of the proposed DGMS method is presented in Algorithm 1.

## D    ADDITIONAL ABLATION EXPERIMENTS

### D.1    ABLATION EXPERIMENTS ON DIFFERENT DATA DISTRIBUTIONS

To evaluate the generalization across different sample distributions, we alternately used samples from the Olympus, Pan, Sony, and Canon camera branches as source domain data for SR network domain generalization training. Table.6 demonstrates that the proposed DGMS method exhibits

**Input:** Source domain LR images $LR$, Source domain HR images $HR$, SR network $M$;
**Output:** Trained SR network $M$;

1  **for** $i \leftarrow 1$ **to** $n$ **do**
2     // Hidden state update regularization
    $SR, \Delta A = M(LR)$;
3     $Loss_{All} = Loss_{SR}(SR, HR) + Loss_{Up}(\Delta A)$;
4     $Loss_{All}.backword()$;
    // Hidden state update regularization + Parameter consistency
        regularization
5     $\theta^a, \theta^b = Random\_sampling(Degradation\_pool)$;
6     $LR^a = De(HR, \theta^a)$; // Degradation operation
7     $LR^b = De(HR, \theta^b)$;
8     $SR^a, \Delta^a A^a, C\Delta BX^a = M(LR^a)$;
9     $SR^b, \Delta^b A^b, C\Delta BX^b = M(LR^b)$;
10    $Loss_{All} = Loss_{SR}(SR^a, HR) + Loss_{SR}(SR^b, HR) + Loss_{Par}(\Delta^a A^a) +$
    $Loss_{Par}(\Delta^b A^b) + Loss_{Consis}(C\Delta BX^a, C\Delta BX^b) + Loss_{Consis}(C\Delta BX^b, C\Delta BX^a)$;
11    $Loss_{All}.backward()$;
12  **end**
13 **return** $M$;

**Algorithm 1:** Overall Training Pipeline

Table 6: Ablation experiments on different data distributions

| Method | Source | Target1 | Target2 | Target3 |
|---|---|---|---|---|
| ORI | Olympus | Pan: 30.30/0.8564 | Sony: 30.39/0.8634 | Canon: 31.03/0.9174 |
| +DGMS | Olympus | Pan: 30.82/0.8535 | Sony: 31.35/0.8785 | Canon: 32.55/0.9240 |
| ORI | Pan | Olympus: 30.50/0.8597 | Sony: 31.45/0.8859 | Canon:31.85/0.9267 |
| +DGMS | Pan | Olympus: 30.84/0.8593 | Sony:31.95/0.8880 | Canon:32.85/0.9272 |
| ORI | Sony | Olympus: 30.66/0.8531 | Pan: 31.15/0.8623 | Canon:32.41/0.9227 |
| +DGMS | Sony | Olympus: 30.71/0.8543 | Pan:31.22/0.8606 | Canon:32.70/0.9278 |
| ORI | Canon | Olympus: 30.48/0.8536 | Pan:30.82/0.8554 | Sony:30.87/0.8763 |
| +DGMS | Canon | Olympus: 30.93/0.8574 | Pan:31.31/0.8601 | Sony:31.79/0.8868 |

strong robustness across diverse data distributions, achieving consistent performance gains on all data branches.

Table 7 presents the performance gains of the proposed DGMS method on the Set5 (Bevilacqua et al., 2012), Set14 (Zeyde et al., 2010), B100 (Martin et al., 2001), Urban100 (Huang et al., 2015), Manga109 (Matsui et al., 2017), and DIV2K (Timofte et al., 2017) datasets, respectively. The proposed method demonstrates significant performance improvements across all data branches, validating its generalization capability across diverse data distributions.

### D.2  ABLATION EXPERIMENTS ON DIFFERENT HIDDEN STATE UPDATE REGULARIZATION LAYERS

We conducted experiments to evaluate the impact of applying regularization terms at different network layers. As demonstrated in Table.8, optimal performance is achieved when the parameter consistency regularization is implemented at the (0, 1-0,1,2,3) layers of the network.

### D.3  ABLATION EXPERIMENTS ON DIFFERENT PARAMETER CONSISTENCY REGULARIZATION LAYERS

We verified the impact of applying parameter consistency regularization constraints at different network layers on SR network performance. From Table.9, it can be found that implementing parameter consistency regularization constraints in the deep layers of the network achieved optimal performance.

Table 7: Ablation experiments on different data distributions

| Method | B100 | Manga109 | Set14 |
|--------|------|----------|-------|
| ORI | 23.6560/0.6565 | 22.7882/0.8067 | 23.1563/0.6769 |
| +DGMS | 25.9315/0.6610 | 25.3353/0.8191 | 25.8956/0.6958 |
| Method | Set5 | Urban100 | DIV2K |
| ORI | 24.4840/0.7645 | 21.2945/0.6690 | 21.5328/0.6052 |
| +DGMS | 28.0974/0.8222 | 23.6006/0.6952 | 24.6743/0.7152 |

Table 8: Ablation experiments on different regularization layers. The Mamba network comprises 6 module groups, each containing 6 layers. In the table, $i, j - k$ indicates that the $k - th$ layer of the $i - th$ and $j - th$ module groups employs the proposed regularization term.

| Method | Pan | Sony | Canon |
|--------|-----|------|-------|
| 0,1,2,3,4,5-0 | 31.23/0.8626 | 31.42/0.8795 | 32.60/0.9266 |
| 0,1,2,3,4,5-5 | 31.32/0.8609 | 31.57/0.8821 | 32.69/0.9253 |
| 0,1,2,3-0,1 | 31.24/0.8603 | 31.34/0.8786 | 32.88/0.9267 |
| 0,1-0,1,2,3 | **31.37/0.8635** | **31.81/0.8854** | **33.19**/0.9279 |
| 0,1-0,1,2,3,4 | 31.26/0.8611 | 31.71/0.8834 | 32.87/**0.9285** |
| 0,1-0,1,2 | 31.20/0.8610 | 31.47/0.8808 | 32.80/0.9269 |
| 0,1,2-0,1,2,3 | 31.24/0.8617 | 31.72/0.8837 | 32.74/0.9264 |
| 0-0,1,2,3,4,5 | 31.24/0.8618 | 31.56/0.8813 | 32.96/0.9278 |
| 5-0,1,2,3,4,5 | 31.17/0.8623 | 31.41/0.8807 | 32.60/0.9249 |

Table 9: Ablation experiments on different parameter consistency regularization layers. The Mamba network comprises 6 module groups, each containing 6 layers. In the table, $i, j - k$ indicates that the $k - th$ layer of the $i - th$ and $j - th$ module groups employs the proposed regularization term.

| Method | Pan | Sony | Canon |
|--------|-----|------|-------|
| 0-0 | 31.25/0.8624 | 31.57/0.8825 | 32.93/0.9262 |
| 0-1 | 31.19/0.8614 | 31.56/0.8815 | 32.67/0.9270 |
| 0-0,1 | 31.32/0.8631 | 31.74/0.8840 | 33.06/0.9274 |
| 3-0 | 31.35/0.8630 | 31.71/0.8834 | 32.94/0.9275 |
| 5-4 | 31.34/0.8640 | 31.50/0.8807 | 32.83/0.9258 |
| 5-5 | **31.37/0.8635** | **31.81/0.8854** | **33.19**/0.9279 |
| 5-4,5 | 31.26/0.8620 | 31.73/0.8840 | 32.84/**0.9281** |

## E  VISUALIZATION RESULTS

As shown in Figure.4, we compared the proposed DGMS with other domain generalization methods. DGMS demonstrated superior detail restoration and noise suppression.

## F  STATEMENT ON LLM USAGE

We used a Large Language Model (LLM), specifically ChatGPT, solely for language polishing and improving the readability of the manuscript. The LLM was not used to generate ideas, conduct experiments, analyze results, or contribute to the research methodology. All scientific content, including the conceptualization, design, implementation, and validation of the work, was entirely carried out by the authors.

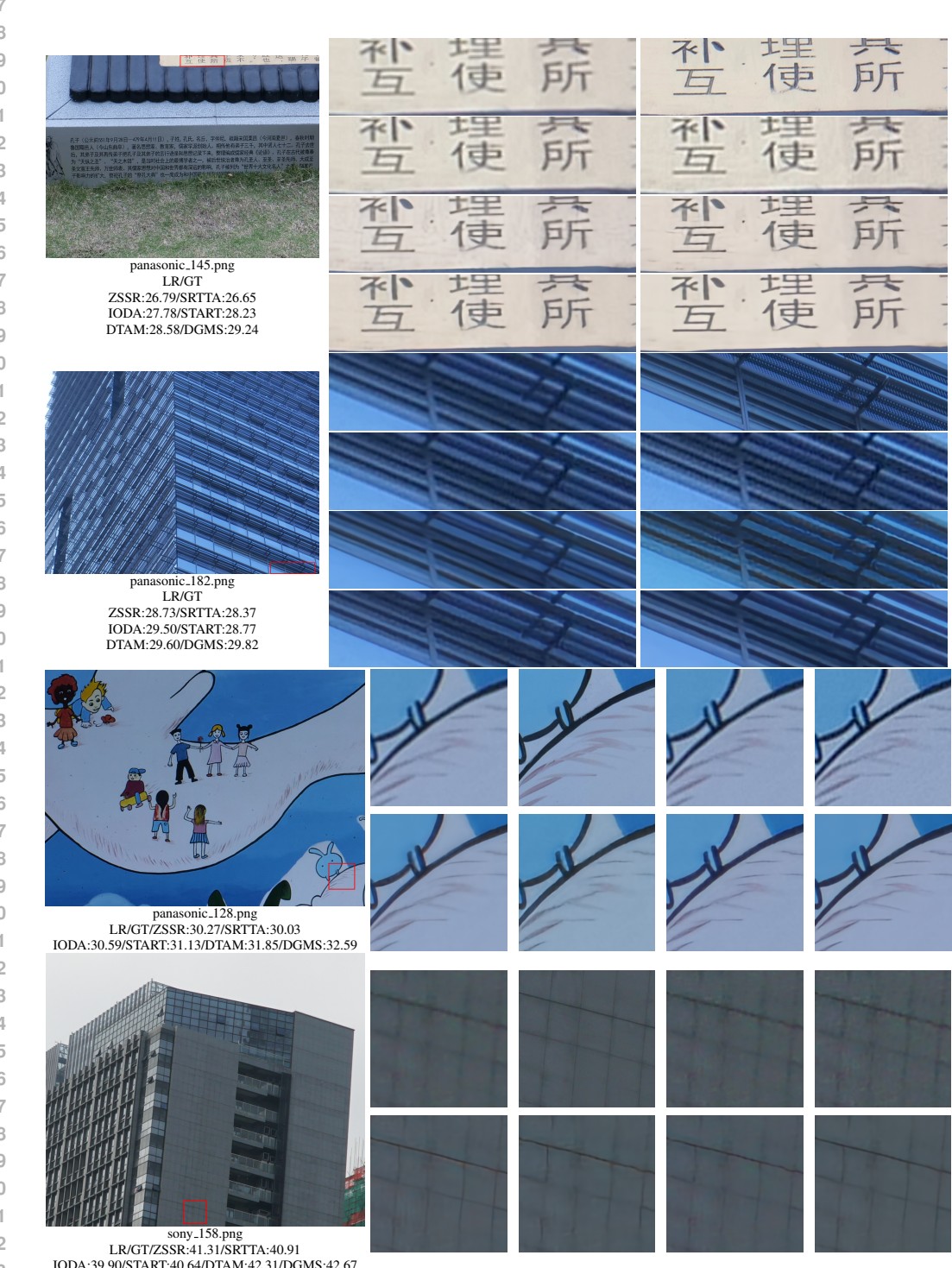

Figure 4: Visual comparison. The large image on the left is the LR image, and the sub-images on the right are LR, GT, ZSSR(Shocher et al., 2018), SRTTA(Deng et al., 2023) ,IODA(Tang & Yang, 2024), START (Guo et al., 2024c) , DTAM (Huang et al., 2024a), MambaIR + DGMS. Please zoom-in on screen.

