# OpenReview forum: "DGMS: Domain Generalization Method for Mamba-based Super-Resolution Networks"
_ICLR.cc/2026/Conference — ICLR 2026 Conference Withdrawn Submission_

### Official Review · Reviewer_YNZ4 · 2025-10-27

**Soundness:** 2
**Presentation:** 1
**Contribution:** 1
**Rating:** 2
**Confidence:** 5

**Summary:**

This paper introduces DGMS, a domain generalization method for Mamba-based super-resolution (SR) networks. Unlike previous approaches focused on high-level vision tasks, DGMS addresses domain generalization in low-level vision by proposing a domain shift metric specific to SR and identifying key variables affecting domain shifts. The method applies hidden state update regularization and parameter consistency regularization to explicitly constrain these variables, thereby improving generalization across various degradation models. Extensive experiments demonstrate that DGMS achieves superior performance over existing state-of-the-art Mamba-based domain generalization methods on diverse datasets and architectures.

**Strengths:**

The proposed method is simple and easy to understand.

**Weaknesses:**

1. The presentation of this paper is very poor; the figures and tables are extremely disorganized and do not meet the standards of a top-tier conference. There is no doubt that it should be rejected.

2. The proposed method lacks any innovation and is merely a straightforward combination of existing works.

3. The real-world datasets evaluated in the paper are too limited. More commonly used real-world datasets, such as RealSR and RealLQ250, should be included in the evaluation.

**Questions:**

na

---

### Official Review · Reviewer_TsgA · 2025-10-30

**Soundness:** 2
**Presentation:** 1
**Contribution:** 2
**Rating:** 2
**Confidence:** 4

**Summary:**

This paper introduces DGMS, which aims to address the performance degradation caused by domain shifts in super-resolution (SR) tasks. The proposed method focuses on improving the generalization capabilities of Mamba-based SR networks by introducing 1) A domain shift metric specifically designed for SR tasks to identify key variables that dominate domain shifts. 2) Two novel regularization terms: Hidden State Update Regularization (HUR) and Parameter Consistency Regularization (PCR), which explicitly constrain the identified key variables to reduce domain shift accumulation and enhance network robustness. 3) A plug-and-play design that is architecture-agnostic and applicable to various Mamba-based SR networks.

Experiments demonstrate the effectiveness of DGMS across multiple datasets, network architectures, and degradation models, achieving superior performance over state-of-the-art domain generalization methods.

**Strengths:**

- The paper is the first to explore domain generalization for Mamba-based SR networks, introducing a domain shift metric and new regularization techniques tailored for SR tasks.

- The authors validate their method across diverse datasets and architectures (MambaIR, MMA, Mair), showcasing its robustness and versatility.

- The proposed DGMS method is designed as a plug-and-play module, making it easy to integrate into existing Mamba-based SR frameworks without requiring significant modifications.

**Weaknesses:**

- Writing and Formatting Issues

The paper's writing and layout require significant improvement to enhance clarity and professionalism.

1) Abstract: The placeholder "***" for the code repository is unprofessional.

2) Formula Presentation: The use of colorful highlights in equations is distracting and unnecessary in an academic paper.

3) Excessive Use of Lists: Overuse of itemized lists in the main text disrupts the flow of the narrative.

4) Table Layout: Table 1 has large amounts of blank space on both sides, and the second row could easily fit into the first row.

5) Related Work Placement: The paper is only nine pages long and has unused space, so the "Related Work" section could have been integrated into the main content instead of the appendix.

6) Figure 3 Inconsistency: In the first row of Figure 3, the middle image has a different height, which disrupts the visual alignment and looks unprofessional.

- Lack of Justification for Mamba-Specific Focus

The paper does not sufficiently justify why the method is specifically designed for Mamba-based SR networks.

1) Limited Relevance: MambaSR is not a dominant architecture in the field of super-resolution, so the focus on Mamba networks may limit the broader applicability of the method.

2) Generalization Potential: If the method aims to address domain shift, it would be more impactful to develop a solution that is generalizable to a wider range of SR architectures, such as CNNs or Transformers. The authors need to provide stronger reasoning for targeting MambaSR specifically.

**Questions:**

see above

---

### Official Review · Reviewer_bxW8 · 2025-10-30

**Soundness:** 3
**Presentation:** 4
**Contribution:** 3
**Rating:** 4
**Confidence:** 5

**Summary:**

This paper introduces a domain generalization approach named DGMS, specifically designed for Mamba-based super-resolution networks. The method aims to enhance model generalization capability on images with unknown degradation distributions by proposing two regularization techniques: hidden state update regularization (HUR) and parameter consistency regularization (PCR). This paper identify key variables governing domain shifts through a dedicated metric and apply explicit constraints to mitigate performance degradation caused by distribution discrepancies between source and target domains.

**Strengths:**

1、First application of domain generalization to Mamba-based low-level vision tasks, achieving expressive results.
2、Novelty:The proposed Hidden state Update Regularization (HUR) and Parameter Consistency Regularization (PCR) significantly improve model generalization performance.
3、Expression is good: Rigorous theoretical derivation and well-designed experiments enhance readability.

**Weaknesses:**

1、Potential limitation in long-range modeling:
In the Hidden State Update Regularization section (lines 236-238), the constraint on discretization matrix A (Eq. 9) aims to reduce the domain shift term. However, in low-level vision tasks (e.g., super-resolution), pixel-level interactions and long-sequence modeling are critical. Since matrix A in state-space models primarily prevents catastrophic forgetting in long sequences, explicitly constraining it may impair Mamba’s ability to capture long-range dependencies, potentially degrading pixel-level perception. The authors should clarify whether this trade-off between domain generalization and spatial modeling capability is justified.
2、Potential interdependence between regularization terms:
The proposed HUR and PCR modules independently constrain the red term and blue term derived from the domain shift formulation. However, the interaction between these terms is not addressed—applying constraints to one term may inadvertently affect the other (e.g., altering the red term via matrix A regularization could propagate changes to the blue term). This decoupled approach lacks justification for its independence assumption and may lead to suboptimal or unstable optimization. The authors should analyze the coupling effects between these terms or provide empirical/theoretical evidence supporting their separability.
3、

**Questions:**

1、Please focus primarily on answering the two questions in the "Weaknesses" section.
2、If possible, please also address the impact of adding these two loss functions on training time. For example, how much would the training time increase for MambaIR when these loss functions are added?
3、Since this paper is related to low-level generalization, it would be more convincing if results from other low-level vision domains (e.g., denoising, deblurring) could be demonstrated.
4、The paper does not clarify whether different weights are assigned to the proposed multiple loss functions.

---

### Official Review · Reviewer_XjMp · 2025-10-31

**Soundness:** 2
**Presentation:** 1
**Contribution:** 2
**Rating:** 2
**Confidence:** 4

**Summary:**

This paper proposes a specialized domain generalization (DG) method, DGMS, to address the performance degradation of Mamba-based Super-Resolution (SR) networks when facing domain shift. The method identifies two key internal variables responsible for shift accumulation and introduces two novel regularization terms, which function as plug-and-play modules. Experimental results validate that DGMS significantly outperforms existing Mamba-based DG methods in cross-domain evaluations.

**Strengths:**

·  Clear and Significant Motivation: The paper addresses the critical problem of domain generalization (DG) for Mamba-based models in low-level vision. It clearly articulates the fundamental incompatibility and discrepancy of applying existing Mamba DG methods—which are designed for high-level vision tasks—directly to low-level counterparts.
·  Simple and Effective Method: The paper proposes an elegant, effective, and plug-and-play method. This approach is theoretically grounded, stemming from a principled analysis of the domain shift accumulation phenomenon, which directly informs the algorithmic design.
·  Comprehensive and Solid Experimentation: The effectiveness of the proposed method is substantiated through thorough and rigorous experimental validation.  A comprehensive set of experiments is presented to demonstrate the method's efficacy across various benchmarks and architectures.

**Weaknesses:**

·  Flawed Theoretical Foundation: A central contradiction exists in the paper. The motivation hinges on the fundamental differences between high-level (patch-wise) and low-level (pixel-wise) tasks , yet the core theoretical framework (Eq. 6) is directly adopted ("followed") from a patch-wise, high-level vision paper (Guo et al., 2024c) . The paper fails to justify why this framework is applicable, nor does it elaborate on the essential differences between these tasks that this borrowed theory might overlook.
·  Overstated Theoretical Novelty: The paper's claim to theoretical innovation appears exaggerated. The core theoretical analysis (Eq. 6 and its proof in Appendix B) is directly adopted from Guo et al. (2024c) . The paper does not extend or adapt this framework to specifically analyze the distinct properties of the low-level SR task, thus limiting its conceptual contribution.
 ·  Poor Justification for the LossUp​ Regularizer: The derivation of the LossUp​ term is not rigorous and lacks justification. The paper fails to explain the significant simplification from the theoretically-derived, input-dependent term (SΔ​(x)A) to a static L1 regularization on the parameters ΔA. Furthermore, the choice of the L1 norm (implying sparsity) over other regularizers (e.g., L2 norm) is arbitrary and lacks theoretical or empirical support.
 ·  Presentation and Formatting Issues: The manuscript contains several presentation flaws. Notably, the abstract concludes with a non-functional placeholder for the code repository ("code is available at ***"). Additionally, the placement of figures and tables is suboptimal (e.g., Figure 1 on page 2).
 ·  Insufficient Analysis of Method Dynamics: The paper lacks a deeper analysis of the proposed method's behavior. The authors do not provide loss curves to illustrate the training dynamics, such as the interplay between the regularization terms (LossUp​, LossConsis​) and the primary reconstruction loss (LossSR​). Consequently, there is no analysis of the potential trade-offs involved in this multi-objective optimization.
 ·  Missing Comparison with Transformer-based Methods: The experimental evaluation is confined to Mamba-based architectures. To properly contextualize the performance and demonstrate the broader utility of DGMS, the comparison should be extended to include strong Transformer-based baselines augmented with relevant domain generalization techniques.

**Questions:**

See  Weakness

---

### Note · Authors · 2025-11-26

I have read and agree with the venue's withdrawal policy on behalf of myself and my co-authors.